# Technological Changes in Wheat-Based Breads Enriched with Hemp Seed Press Cakes and Hemp Seed Grit

**DOI:** 10.3390/molecules27061840

**Published:** 2022-03-11

**Authors:** Verena Wiedemair, Kathrin Gruber, Nataly Knöpfle, Katrin E. Bach

**Affiliations:** 1Department of Food Technology and Nutrition, MCI, Maximilianstr. 2, 6020 Innsbruck, Austria; verena.wiedemair@mci.edu (V.W.); nataly.knoepfle@mci.edu (N.K.); 2DIETZ Consulting e.U., Werndlgasse 1b, 4523 Neuzeug, Austria; k.gruber@dietz-consulting.at

**Keywords:** hemp, bread, componential changes and interactions, by-product upcycling, texture profile analysis, colour analysis, principal component analysis

## Abstract

Hemp and hemp seed press cake—a by-product of hemp oil production—are high-protein, gluten-free raw materials that are often used to enhance the nutritional value of breads. The addition of hemp materials, however, often negatively impacts the technological parameters of breads. Consequently, this study investigated whether and how much the addition of various by-products of hemp seed press cakes to wheat bread mixtures adversely affects the texture and colour profile. The texture profile and colour were determined using a texture analyser and tristimulus measurements. The particle size of raw materials was also measured. Principal component analysis was then used to visualise the correlation between all measured values as well as nutritional parameters. The results showed that the addition of only 1% of some hemp raw materials caused significant technological changes (*p* > 0.05). Hemp raw materials increased bread hardness and decreased elasticity. The colour of breads containing 1% hemp was also visibly darker than the reference bread. The addition of more hemp led to further darkening and the deterioration of the technological parameters of the products. Consequently, while various hemp materials have high nutritional value, a balance with sensory properties, e.g., textural and colour, has to be reached.

## 1. Introduction

Bread is one of the oldest food products and a staple of the human diet in many countries [1,2]. In recent years, food-related health concerns have increased consumer demand for functional foods, foods with improved nutritional quality and products with reduced allergens for preventing food and age-related diseases [3,4,5,6,7]. Hemp and its by-products from oil production are often used as additional raw materials in breads because of their high protein content and value, as well as their overall high nutritional value. The protein fraction of hemp is similar to egg white, with a high amount of arginine [7,8,9]. Furthermore, hemp is a gluten-free raw material, which is why it is also often added to gluten-free dough mixtures [10].

A drawback of hemp, however, is that the lack of gluten greatly impacts the texture and baking properties of bread. Studies have shown that consumer acceptance of bread greatly depends on textural properties and that as little as 5% hemp flour alters the texture profile of breads [10,11].

In addition to hemp flour and hemp seed (*Cannabis sativa*) press cakes, which are a by-product of hemp oilseed production, beans and their by-products, vegetables and their by-products, mushrooms and groundnuts, as well as other seeds, such as flaxseed, are also often used to enhance the nutritional value of gluten-containing and gluten-free breads [12,13,14,15,16,17]. Vegetables and their by-products are usually used to enhance the amount of bioactive compounds and mineral content as well as to add fibre. A study on the addition of broccoli leaf powder to gluten-free bread showed that the overall nutritional value, as well as technological parameters such as bake loss, can be improved by the addition of non-traditional bread ingredients [15]. Other studies highlighted that the addition of ground nuts can improve the sensory properties of breads, but technological parameters, such as bread volume, are often adversely affected [14,18]. Like hemp, beans are often used to increase fibre content and enhance protein composition [12,13,19,20]. Hemp seed cakes have also proven to be a valuable source of protein, dietary fibre and micronutrients, such as antioxidants and vitamins [19,20,21,22].

In the European Union, foods with high nutritional value can be labelled with nutrition and health claims according to Regulation (EC) No 1924/2006. One such claim is high protein, which indicates that at least 20% of the energy of the product is provided by protein [23].

Therefore, the current study investigated breads with 1% of various hemp raw materials made from press cakes and hemp grit, as well as breads with higher amounts of hemp materials, to meet the requirements of the aforementioned health claim. Previous studies have focused on hemp flour or hemp seed press cake and did not investigate the impact of hemp seed grit or hemp press cake isolate on technological parameters. Consequently, this study will help to better understand how different hemp by-products can be used and implemented in breads.

Therefore, small amounts of each hemp raw material were used to study whether and to what extent the technological properties of the finished product are impacted by minimal changes. A previous study showed that 5% hemp raw material has an effect on technological parameters [11]. Therefore, to investigate whether even lower amounts also have an effect, only 1% hemp material was added. Then, to meet the criteria for the EU health claim of high protein, hemp by-products were added in varying amounts according to their protein content to study how technological properties are affected. The necessary amounts of each hemp raw material were determined according to information provided by the supplier of the raw material. Consequently, the results of the current study, on the one hand, provide insight into the feasibility of high-protein breads containing hemp in an industrial context and, on the other hand, are essential to better understand how various hemp by-products influence certain bread properties, which often correlate with consumer acceptance.

## 2. Results

### 2.1. Characterisation of Raw Materials

The dry matter (DM) of the raw materials ranged from 92.6 to 96.8% (ΔDM = 4.2%) with a mean of 94.1 and a median of 93.8%. The standard deviation (SD) and relative standard deviation (RSD) were 1.7 and 1.8%, respectively. The water-binding capacity (WBC) ranged from 1.2 to 1.9% (ΔWBC = 0.7%) with a mean and median of 1.4%. SD and RSD were calculated to be 0.3 and 21%, respectively. The high RSD value stems from the fact that the WBC of HSG is quite high compared to all other raw materials. The least gelation concentration (LGC) varied between 14 and 22% (ΔLGC = 8%). The means of all mentioned parameters are summarised in Table 1 for all raw materials. SDs are also provided, if applicable.

The means and medians of the presented parameters are similar, which indicates that the data are not highly skewed. All raw materials had a dry matter content of over 90%. The WBCs of all but one hemp raw material (HP54) differed significantly from that of the wheat flour mixture (*p* < 0.05). The LGCs of samples also differed, with the half-and-half mixture of wheat and whole wheat flour (WFM) and hemp press cake flour protein (46%) (HP46) yielding the lowest value.

The distributions of particle sizes of all raw materials are shown in Figure 1, and Table 1 lists the first and last deciles as well as the medians for all samples. Measurements were performed at 1 bar with a 90% feed rate on a Mastersizer 2000 (Malvern Panalytical GmbH, Kassel, Germany). Hemp grit comprises larger particles, whereas hemp protein from dehulled seeds (hemp press cake flour protein (54%) (HP54)) is composed of rather small particles. Table 1 also shows that HP46 and HPF have a similar particle size, which is due to the fact that both are produced from press cakes of whole hemp seeds. The median particle size of WFM is slightly smaller than the values for HP46 and hemp press cake flour (HPF), because WFM comprises wheat and whole wheat flour. Furthermore, the half-and-half mixture of wheat flours has the widest relative range, which is to be expected since flours with different particle sizes were mixed. Consequently, the maximum particle size, which is approximately 100 µm, represents wheat flour, and the remaining larger particles are attributable to whole wheat flour, where particle sizes are generally larger. Figure 1a highlights that all raw materials except HP46 comprise particles larger than 1250 µm, which could not be measured. Figure 1b further shows that WFM and hemp seed grit (HSG) contain more particles with sizes larger than 1250 µm compared to HPF and HP54. Hemp seed grit shows the smallest relative particle size range (RR) values. However, HSG comprises many particles with a size of over 1250 µm, which is why the RR value is only an estimate and based on the assumption that the particle size distribution is symmetrical. Consequently, the RR value for HSG is not representative of the whole sample. The second smallest RR value was calculated for HP46, for which the particle size of the whole sample could be measured. The particle size distributions of WFM, HP54 and HPF all show at least two maxima, which is also reflected in higher RR values.

The colours of the raw materials ranged from beige to brown. All samples had positive a* and b* values. The mixture of wheat flours (WFM) showed the lightest colour, whereas HP46 had the darkest colour. HP46 was also the sample with the greatest ΔE* relative to the wheat mixture reference. HP54 had the highest a* and b* values. All L*, a*, b* and ΔE* values are listed in Table 2.

All calculated ΔE* values are over 20. As a rule of thumb, samples are perceived as a different colour if the ΔE* value is greater than 5 [24]. Consequently, the raw materials can be easily recognised by their colour.

A scaled and centred principal component analysis (PCA) was performed for explorative data analysis. Figure 2a shows that, together, the first and second principal components (PC) explain 81.5% of the variance in the data set. Additionally, Figure 2b highlights that only the first three components have an eigenvalue over 1. Since the data set is quite small, it was expected that a high amount of variance would be explained by a few PCs. The loadings, which are represented by the red arrows in Figure 2, further show that a*, b*, fat, energy, d(0.5), LGC and WBC have a high correlation with PC1. This is also supported by the loadings (see Appendix A). DM and DF have smaller errors, which is related to the lower loadings for PC1. The loadings for PC2 show that protein, RR, carbohydrates and L* have a strong impact on PC2. Dry matter and dietary fibre have high loadings for PC3.

Figure 2a further shows that WFM is the only sample with a negative value on PC2, a component closely related to carbohydrate content. This is unsurprising, as WFM contains over 50 g/100 g carbohydrates, whereas all remaining samples have less than 10 g/100 g (see Section 4). The hemp raw materials mostly line up from largest to smallest d(0.5) in PC1. Additionally, it is interesting to note that HP46 and HPF are more closely grouped together, most likely because they have a similar particle size and are both produced from whole hemp seeds, which is also reflected in the smaller a* and b* values of these raw materials. HP54 is produced from dehulled hemp, which is why its colour is distinctly different. Figure 2a also confirms that HSG is different from the other hemp materials since it is produced from whole seeds, not from press cakes. Consequently, HSG has a much higher fat content (46 g/100 g compared to less than 10 g/100 g for all other hemp raw materials) and thus also contains more energy (2380 kJ/100 g compared to less than 1400 kJ/100 g for all other hemp raw materials (see Section 4)). HSG also has the highest WBC, although the difference is not as pronounced. This is also the reason why the loading arrows for WBC, energy, fat content and d(0.5) point to this sample. Differences in the remaining hemp raw materials (HPF, HP46 and HP54) mostly stem from different colours (a* and b* values) and protein contents.

### 2.2. Characterisation of Doughs and Breads

Doughs and breads were freshly prepared as outlined in Material and Methods, Section 4.1 and Section 4.2. Sample doughs with 1% hemp raw material expanded by between 63 and 70 mL in 110 min. The reference dough and dough with hemp seed grit expanded the most (70 mL), whereas HP46-1-D expanded the least (63 mL). HPF-1-D and HP54-1-D expanded by 68 mL and 67 mL, respectively. The difference in expansion between dough with 1% hemp and the reference is not significant (*p* > 0.05).

Doughs with higher amounts of hemp expanded significantly less than the reference dough (*p* < 0.05). HP46-14-D expanded the least (50 mL) and H54-10-D expanded the most (53 mL). HPF-18-D expanded by 52 mL.

The colours of doughs with 1% hemp raw material were all similar, which can also be concluded from Table 2, where L*, a*, b* and ΔE* values are listed. All ΔE* values are below 3, which also suggests that the dough colours are very similar. Depending on the product, a ΔE* value above 2 or 3 can be perceived by consumers [24,25,26]. Doughs with more than 1% hemp raw material all show a ΔE* value above 5, which indicates that the samples have a distinctly different colour.

The colours of the crumbs of all breads with 1% hemp raw materials were similar (see Table 2). All a* and b* values are positive, and all ΔE* values are below 5. Samples with ΔE* > 5 are perceived as different colours by consumers [24]. This is the case for all breads with higher amounts of hemp material (see Table 2). The bread samples HSG-1-B and HP54-1-B have ΔE* values of less than 2, which suggests that there is only a very slight colour difference, which is likely not perceived by consumers. HPF-1-B, HP46-1-B and HP54-10-B show ΔE* values close to 5, which suggests that consumers are likely to recognise the colour difference. However, as seen in Figure 3, HP54-10-B is similar in colour to breads with only 1% hemp raw material, despite the sample containing 10% hemp raw material. Figure 4 further illustrates that HPF-18-B and HP46-14-B are clearly darker than the other samples.

The results of the colour analyses show that breads with HSG-1-B and HP54-1-B are the most similar to the reference bread.

The results of texture analysis are listed in Table 3. Dough stickiness ranges from −0.26 to −0.70 N (ΔStickiness = 0.44 N), with HSG-1-D having the lowest value and HP46-14-D having the highest. The mean and median stickiness values were −0.53 and −0.55 N, respectively. SD and RSD values were calculated to be 0.16 N and 31.0%, which suggests great variation between samples. ANOVA further revealed that all doughs containing a hemp raw material, except for HP46-1-D, differed significantly from the reference dough regarding their stickiness (*p* < 0.05).

For the texture analysis of breads, hardness was measured, and cohesion, elasticity, gumminess and chewiness were calculated (see Table 3). Hardness ranged from 5.84 to 15.71 N (ΔHardness = 9.88 N), with WFM-B having the softest crumb and HP46-14-B having the hardest. The median and mean of all samples were 8.21 and 10.00 N, respectively. For SD and RSD, values of 3.99 N and 39.9% were calculated. Cohesion values are usually between 0 and 1 but ranged from 0.89 to 1.07 in the presented measurements. Cohesion corresponds to the quotient of the second and first peak areas. Consequently, a cohesion value over 1 means that the second peak area is larger than the first. This indicates that the crumb of the reference bread fully springs back into place after the first penetration. This is in agreement with a previous study, which also showed that freshly baked wheat bread springs back into place after the first compression [27]. When adding as little as 1% hemp raw material, this effect significantly decreased (*p* < 0.05). The cohesion of HPF-18-B and HP46-14-B also significantly differed from that of WFM (*p* < 0.05). HP54-10 did not differ significantly from WFM (*p* > 0.05). HP54-10 had the highest cohesion but also the highest standard deviation. The median and mean cohesion of all breads had values of 0.94 and 0.96 with SD and RSD values of 0.07 and 7.1%, respectively. Elasticity ranged from 1.05 to 1.13 (ΔElasticity = 0.08) and had a median and mean of 1.07 and 1.08, respectively. SD and RSD were both quite low, with values of 0.03 and 2.6%. Gumminess and chewiness ranged from 5.99 to 14.44 (ΔGumminess = 8.45 N) and 6.28 to 15.91 N (ΔChewiness = 9.62 N), respectively. Mean and median gumminess values of all samples were 9.42 and 8.11 N, and SD and RSD were 3.30 N and 35.1%. The mean chewiness of all samples was 10.19 N, and the median value of the same parameter was 8.88 N. SD was calculated to be 3.67 N with an RSD value of 36.0%, which indicates a high variance between samples.

The mean and median cohesion and elasticity over all samples of breadcrumbs were close. Regarding hardness, gumminess and chewiness, the means and medians were further apart, mostly because HP54-1-B, HPF-18-B and HP46-14-B exhibited higher values than those of the remaining samples. The cohesion of all breads containing hemp, except for HP54-10-B, differed significantly from that of the reference (*p* < 0.05). Regarding elasticity, all samples except for HPF-18-B also differed significantly from the reference (*p* < 0.05). Consequently, even the addition of as little as 1% hemp raw materials caused changes in technological properties.

The hardness and chewiness of bread containing 1% HP46 did not differ significantly from those of the reference bread (*p* > 0.05). All other breads showed significant differences in these parameters compared to the reference bread (*p* < 0.05). Gumminess differed significantly from the reference bread for all samples (*p* < 0.05). In the present study, the data for dough and breads are in good agreement. The presented data also suggest that HP54-10-B had similar textural properties to samples with 1% hemp materials, which is interesting considering the large increase in hemp material.

Scaled and centred PCAs were computed to perform explorative data analysis (see Figure 4). The dough colours were very similar (see Figure 4a), which is why the samples are located close to each other. Raw materials, on the other hand, had very different colours, as discussed in the previous section. Breads and doughs mostly differed in their luminosity (L*), which is reflected in PC2. This is also reflected in PC loadings. The parameters a* and b* show high loadings for PC1, whereas L* has high loadings for PC2 (see Appendix A)

Figure 4b shows that hardness, gumminess and chewiness are closely related. This is to be expected because gumminess is calculated by multiplying hardness and cohesion, and chewiness is the product of elasticity and gumminess. These three parameters also have the highest and similar loadings for principal component 1 (see Appendix A). Elasticity and stickiness have the lowest loadings for PC1. At first glance, elasticity and stickiness appear to be closely related as well. However, the loadings reveal that only cohesion and elasticity have high loadings for PC2. This is also shown in Figure 4, as the loading arrow for stickiness is rather short. As expected, hardness, gumminess and chewiness have the lowest loadings for PC2. Lastly, stickiness has high loadings for PC3. Figure 4b also suggests that adding 1% HP54, 18% HPF or 14% HP46 as a raw material yields distinctly different texture properties compared to the reference bread. ANOVA revealed that bread with 1% HP46 as a hemp raw material is the most similar to the reference bread, since their hardness, gumminess and chewiness do not differ significantly. This is also shown in the PCA, where the mentioned parameters are mostly represented in PC1, and bread with 1% HP46 can be found furthest on the right-hand side of the plots for all breads with hemp raw materials. Interestingly, HP54-10 has a similar hardness, chewiness and gumminess to the reference, despite the higher amount of hemp raw material. It is likely that since HP54% contains mainly protein, textural properties are not affected as much.

## 3. Discussion

### 3.1. Characterisation of Raw Materials

Table 1 presents various parameters of all raw materials. It highlights that raw materials were highly similar in dry matter content but differed in water-binding capacity (WBC) and least gelation concentration (LGC). Both parameters have an influence on the technological properties of doughs and breads. WFM and HP46 yielded the lowest LGC values, which is most likely due to the fact that wheat flour has a high amount of starch (over 50% [28]). According to Schultz et al., this leads to good gelling properties [29]. Hemp, on the other hand, contains only about 2% starch, which is why its gelling properties mainly stem from proteins [29]. HSG is prepared similarly to WFM by grinding seeds, which might explain why its LGC is close to the LGC of WFM. HPF, HP46 and HP54 are all prepared from press cakes. The high LGC of HP54 is most likely due to a high number of protein aggregates in the sample, which prevents gelation processes and was also observed in another study. [30]. HPF has a high amount of dietary fibre (see Table 4), which does not contribute to gelling and could consequently explain why the LGC of HPF is higher than the LGC of HP46.

WBC is influenced by particle size. For wheat, it is generally expected that finer flours have a higher WBC [31]. However, there are also studies that conflict with this expectation and show that coarse wheat flour exhibits a higher WBC because it retains more water-soluble compounds than finer flour. WBC is also influenced by the content and types of proteins present in the raw materials [32]. The data of the present study suggest that for hemp raw materials, the particle size positively correlates with WBC. This might be due to factors such as the type of protein but also the amount of fat (see Table 4). Consequently, an analysis of various protein fractions would be beneficial to gain a better understanding of the driving factor for WBC in each raw material.

### 3.2. Characterisation of Doughs and Breads

Dough expansion is an indicator of the textural parameters of breads. In the presented study, doughs with higher amounts of hemp expanded significantly less than the reference dough (*p* < 0.05). This finding is in good agreement with other studies, which showed that dough expansion is significantly influenced by the addition of at least 5% hemp flour [20,33]. Regarding doughs with only 1% hemp, it is likely that this effect was not observable due to the small amount. Additionally, the use of dry yeast for baking might have also had a standardizing effect on volume expansion.

Bread colour is an important parameter for consumer acceptance. The presented data suggest that colour differences are already perceivable if only a very small amount of hemp raw material is added. However, studies on breads with insect powder, pea flour and hemp flour showed that consumers favoured darker crumb colours in breads [11,34,35]. It is suggested that the reason for the increased preference is that darker breads resemble whole wheat or multigrain breads, which are generally perceived to have nutritional benefits [11,36]. Another interesting aspect is that colour differences were greater between breads than between doughs. This suggests that hemp is subject to more intense browning reactions during baking.

In general, consumer acceptance regarding colour and technological parameters is a great concern with the addition of high-protein substitutes, such as hemp or insect powder. Studies have found that breads that contained hemp were darker in colour and harder and denser than the hemp-free reference [10,11]. This is in good agreement with the data presented in this study.

Previous studies on gluten-free breads supplemented with hemp flour, however, also suggest that consumer acceptance of texture only decreases if more than 10% of hemp raw material is added [10,11]. Hayward and McSweeney found in their study that gluten-free bread supplemented with only 5% hemp flour had a similar consumer acceptance level to the reference bread regarding all tested sensory attributes [11].

Consequently, it would be interesting to test consumer acceptance of the bread produced for this study and investigate whether or not similar results to those of Hayward and McSweeney could be obtained. It is assumed that among breads with high amounts of hemp raw material, HP54-10 has a high potential for consumer acceptance, as differences in colour and texture are less pronounced. Consequently, consumer testing of this formulation would be interesting.

## 4. Materials and Methods

### 4.1. Raw Materials

All hemp raw materials were provided by Hanfland GmbH (Laa a. d. Thaya, Austria and comprised: hemp seed grit (HSG), hemp press cake flour (HPF), hemp press cake flour protein (54%) (HP54) and hemp press cake flour protein (46%) (HP46). All products were organically grown in Austria and are certified with the EU organic logo [37]. Hemp seed grit was produced by coarsely grinding hemp. All other hemp raw materials were produced from press cakes, which are a by-product from hemp oil production. In short, hemp seeds were cold-pressed to release oil, and the solid residue was ground into a flour to obtain HPF. For HP46 and HP54, the press cake was again ground and then sieved to remove non-protein fractions. The press cakes of HPF and HP46 comprised only whole hemp seeds, whereas the press cake for HP54 was made from dehulled hemp seeds.

Wheat flour (WF) (Austrian flour type W700) and whole wheat flour (WWF) (Austrian flour type W1800) were purchased from Anton Rauch GmbH & Co KG (Hall i. Triol, Austria). A half-and-half mixture (w%) was prepared and served as reference material. In Table 4, important nutritional parameters of all raw materials according to packaging are presented.

### 4.2. Dough Preparation and Bread Making

Hemp raw materials were added to a standard bread dough mixture of wheat flour, whole wheat flour, salt, dry yeast and water. The ratio of wheat flour to whole wheat flour was 1:1 in all doughs, and the amount of water was adapted according to the water-binding capacities of the used hemp raw materials, which were experimentally determined before dough preparation [38]. The composition of all doughs is shown in Table 5. To meet the criteria for the EU health claim of high protein, 20% of the energy has to come from protein. The necessary amounts of each hemp raw material were determined and calculated according to information provided by the supplier of the raw material using the following equation:(1)$$\mathrm{Energy}\;\mathrm{from}\;\mathrm{Protein}\;\mathrm{in}\%=\frac{F\times \left(m_{\mathrm{WFM}}\times P_{\mathrm{WFM}}+m_{H}\times P_{H}\right)}{m_{\mathrm{WFM}}\times E_{\mathrm{WFM}}+m_{H}\times E_{H}}\times 100$$
m_WFM_: Amount of WFM/g/100 g;P_WFM_: Protein content of WFM/g/100 g;W_WFM_: Energy of WFM/kJ;m_H_: Amount of hemp raw material/g/100 g;P_H_: Protein content of hemp raw material/g/100 g;E_H_: Energy of hemp raw material/kJ;F: Factor to convert g protein to kJ (1 g = 17 kJ).

**Table 5 molecules-27-01840-t005:** Ingredients for all prepared doughs/breads. HSG—hemp seed grit; HPF—hemp press cake flour; HP46—hemp press cake flour protein (46%); HP54—hemp press cake flour protein (54%); WFM—half-and-half mixture of wheat and whole wheat flour.

Dough	WFM/%	Dry Yeast/%	Salt/%	Water/%	Respective Hemp Raw Material/%
WFM-D	59.72	1	1.1	38.18	0
HPF-1-D	56.72	1	1.1	40.18	1
HP46-1-D	56.72	1	1.1	40.18	1
HP54-1-D	58.72	1	1.1	38.18	1
HSG-1-D	56.72	1	1.1	40.18	1
HPF-18-D	39.72	1	1.1	40.18	18
HP46-14-D	43.72	1	1.1	40.18	14
HP54-10-D	49.72	1	1.1	38.18	10

For dough preparation, all raw materials were mixed for 5 min at 100 rpm in a conventional food processor (Cromargan, WMF, Geislingen, Germany) and then fermented for 1 h at 30 °C in containers greased with sunflower oil and sprinkled with wheat flour. Lastly, doughs were baked at 195 °C for 30 min in a standard kitchen oven. Figure 3 shows the finished breads, which were cut in half.

### 4.3. Characterisation of Raw Materials

All hemp raw materials and the wheat flour mixture were analysed for their dry matter, particle size, colour, water-binding capacity and least gelation concentration. Dry matter (DM) was analysed according to the standard protocol [39].

Particle size was determined with the Mastersizer 2000 ((Malvern Panalytical GmbH, Kassel, Germany) according to a previously published method [40]. Air pressure was 1 bar with 90% feed rate. The refractive index was assigned a value of 1.530, and absorption was set to 0.100. The relative range (RR) of particle size is calculated as the quotient of the difference between the highest and lowest deciles and the median:(2)$$\mathrm{RR}=\frac{d\left(0.9\right)\;-\;d\left(0.1\right)}{d\left(0.5\right)}$$

Colours of raw materials were determined using a tristimulus colourimeter (CR-5, Konica Minolta Holding, Inc., Marunouchi, Japan). The variables L* (luminosity), a* (red to green) and b* (yellow to blue) were measured, and the colour difference was calculated according to the following equation:(3)$$\Delta E^{*}=\sqrt{{\left(\Delta L^{*}\right)}^{2}+{\left(\Delta a^{*}\right)}^{2}+{\left(\Delta b^{*}\right)}^{2}}$$

Water-binding capacity (WBC) was determined in triplicate according to the method of Sathe et al. as modified by Ionescu et al. [38,41]. Approximately 0.5 g of sample was weighed in a centrifuge tube of known mass, and 8 mL of deionised water was added. The suspension was vortexed for 5 s. After 30 min, samples were centrifuged at 20 °C and 4500 rpm, and the supernatant water was removed. The tube was reweighed, and water-binding capacity was calculated and expressed in %.

Least gelation concentration (LGC) was determined in triplicate using the method of Badar with slight modifications [42]. Suspensions of 10, 12, 14, 16, 18, 20 and 22% of all raw materials in deionised water were prepared and heated for 1 h at 95 °C. Samples were then cooled to 4 °C overnight. LGC was determined visually by assessing the concentration at which the sample did not slip in the inverted tube.

### 4.4. Dough Characterisation

Dough expansion, stickiness and colour were analysed. The latter was measured using a tristimulus colourimeter (CR-5, Konica Minolta Holding, Inc., Marunouchi, Japan), as described in the previous section.

Dough stickiness was determined with a texture analyser (TA.XTplus C Texture Analyser, Stable Micro Systems Ltd., Surrey GU7 1YL, United Kingdom) with a Chen-and-Hoseney cell based on a protocol from Grausgruber et al. [43]. For doughs, an SMSP/25P probe was used, and for breads, an SMSP/25 probe was used. Then, dough was formed into a ball and put into an extrusion cell. A small amount of dough was extruded and wiped off with a spatula. Then, measurement was started with a trigger force of −0.049 N. Test and pre-test speeds were set to 0.5 mm/s, and post-test speed was 10.0 mm/s. Compression force was set to 0.785 N. All samples were measured at least nine times.

### 4.5. Bread Characterisation

The colours of breads were determined as previously described using a tristimulus colourimeter. Breadcrumb hardness was measured using a texture analyser (TA.XTplus C Texture Analyser, Stable Micro Systems Ltd., Surrey GU7 1YL, UK). Cohesiveness, springiness, gumminess and chewiness were subsequently calculated. Then, cylindrical samples with 25 mm diameter and 20 mm height were prepared and analysed according to an adapted method by Kadan et al. [44]. All measurements were performed as double compressions in order to simulate chewing. The trigger force was set to 0.049 N, and the test and post-test speeds were set to 2 mm/s. The pre-test speed was 1 mm/s, and the travel distance was set to 10 mm. All samples were measured six times.

### 4.6. Statistics and Multivariate Data Analysis

Means and standard deviations (SDs) were calculated for all parameters for repeat measurements as well as for the whole sample set. The ratio of SD to mean expressed in percent, also called relative standard deviation (RSD) or coefficient of variation, was calculated for the sample set in order to estimate sample dispersion. Principal component analysis (PCA) was performed in R Stats using the prcom function and was used for explorative data analysis. For all PCA computations, data were scaled and centred. The Suitability of PCA was evaluated using scree plots (see Figure 2 and Appendix A). Additionally, the impact of parameters on certain principal components was evaluated through loadings.

## 5. Conclusions

The present study explored technological changes in breads with 1% of various hemp materials and breads that are high in protein, with 20% of the energy coming from protein. Colour differences between doughs with 1% hemp raw materials compared to the reference material were very small (ΔE* < 2.4); however, colour differences between the remaining samples and the reference were higher, with ΔE* < 20.1. After baking, the colour difference increased for all samples, which suggests that hemp raw materials are subject to more browning reactions. HPF-1-B and HP46-1-B showed a ΔE* > 4, which means that consumers will likely perceive these breads to be darker than the reference bread. HPF-18-B and HP46-14-B showed a ΔE* > 15, which indicates large colour differences, which are visible in Figure 4. Consumers often associate a dark crumb with whole grain or multigrain breads, which is why the visible colour difference might not affect consumer acceptance. However, this still needs to be tested. Additionally, all doughs prepared with 1% hemp material showed similar expansion behaviour. This was expected since only a small amount of wheat flour mixture was substituted, and the addition of dry yeast had a standardizing effect on dough volume expansion. The remaining samples with more hemp material had a significantly smaller volume expansion, which is in accordance with previously published studies [20,33].

The texture of most doughs containing hemp raw material differed significantly from the reference dough. As expected, high dough stickiness translated to a high elasticity in breads. Previous studies showed that consumer acceptance only decreased when replacing 10% of the flour or more with hemp [10,11]. In the current study, however, bread prepared with 10% HP54 showed similar textural properties to the reference. Consequently, this formulation might be acceptable to consumers; however, this still needs to be investigated.

This study proved that even the addition of as little as 1% hemp leads to statistically significant technological changes in breads. The current study further revealed that by carefully selecting hemp raw materials (e.g., hemp proteins), the protein content of breads can be significantly increased, while textural properties remain similar to the reference.

## Figures and Tables

**Figure 1 molecules-27-01840-f001:**
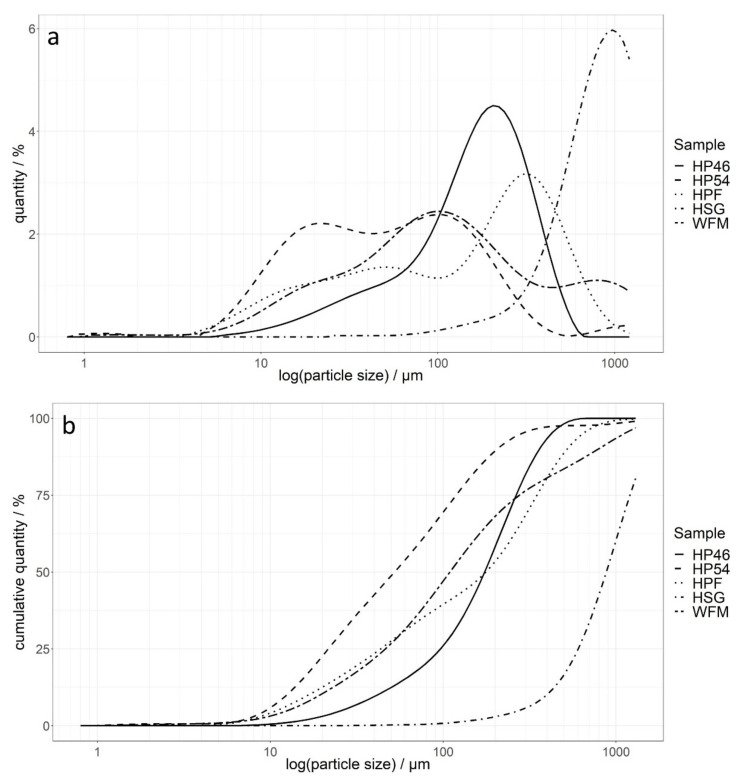
Sample size distribution (**a**) and cumulative sample size distribution (**b**) of all raw materials at 1 bar with 90% feed rate. HSG—hemp seed grit; HPF—hemp press cake flour; HP46—hemp press cake flour protein (46%); HP54—hemp press cake flour protein (54%); WFM—half-and-half mixture of wheat and whole wheat flour.

**Figure 2 molecules-27-01840-f002:**
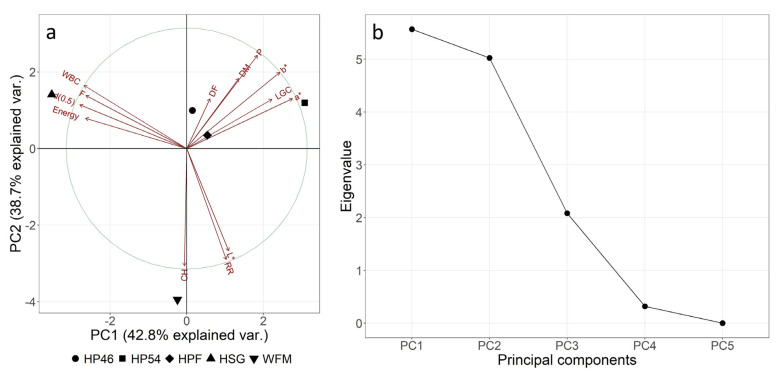
Scaled and centred biplot (**a**) and scree plot (**b**) of principal component analysis of physical (Table 1) and nutritional (Appendix A) parameters of raw materials. HSG—hemp seed grit; HPF—hemp press cake flour; HP46—hemp press cake flour protein (46%); HP54—hemp press cake flour protein (54%); WFM—half-and-half mixture of wheat and whole wheat flour.

**Figure 3 molecules-27-01840-f003:**
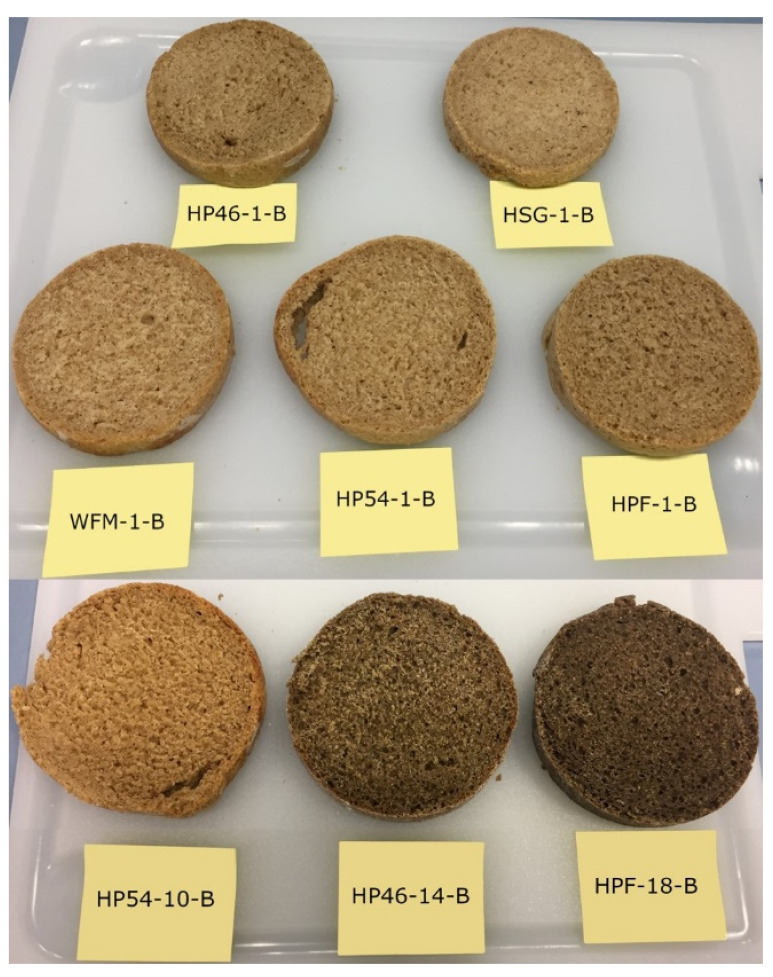
Breads (-B). HSG—hemp seed grit; HPF—hemp press cake flour; HP46—hemp press cake flour protein (46%); HP54—hemp press cake flour protein (54%); WFM—half-and-half mixture of wheat and whole wheat flour.

**Figure 4 molecules-27-01840-f004:**
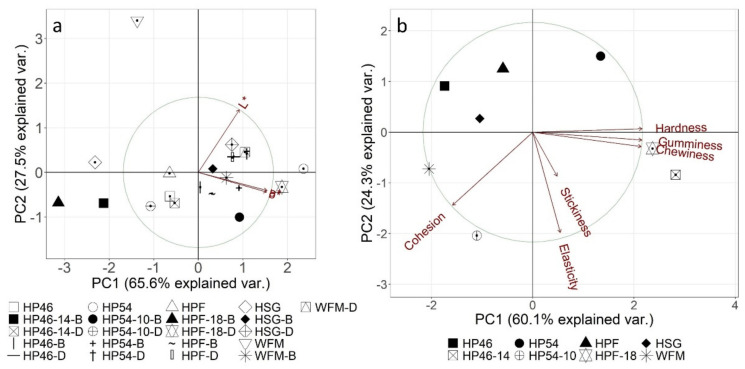
Scaled and centred biplots of colour (**a**) and texture (**b**) of samples. HSG—hemp seed grit; HPF—hemp press cake flour; HP46—hemp press cake flour protein (46%); HP54—hemp press cake flour protein (54%); WFM—half-and-half mixture of wheat and whole wheat flour.

**Table 1 molecules-27-01840-t001:** Means with standard deviations (where applicable) of dry matter (DM), water-binding capacity (WBC) and least gelation concentration (LGC) of hemp seed grit (HSG), hemp press cake flour (HPF), hemp press cake flour protein (46%) (HP46), hemp press cake flour protein (54%) (HP54) and half-and-half mixture of wheat and whole wheat flour (WFM). Median particle size and relative particle size range (RR), as well as first and last deciles, are also reported for all raw materials. WBC and LGC were determined in triplicate.

Raw Material	DM/%	WBC/%	LGC/%	d(0.1)/µm	d(0.5)/µm	d(0.9)/µm	RR
HSG	94.4	1.9 ± 0.4 *	16	397	879	1506	1.3
HPF	92.8	1.4 ± 0.1 *	18	17	177	521	2.9
HP46	93.8	1.5 ± 0.1 *	15	42	174	361	1.8
HP54	96.8	1.2 ± 0.1	22	13	52	205	3.7
WFM	92.6	1.2 ± 0.1	15	19	109	790	7.0

* Samples differ significantly from WFM (*p* < 0.05).

**Table 2 molecules-27-01840-t002:** Colour of all raw materials: doughs (-D) and breads (-B) expressed in CIELAB colour space. L*—luminosity; a*—red to green; b*—yellow to blue; ΔE*—colour difference; HSG—hemp seed grit; HPF—hemp press cake flour; HP46—hemp press cake flour protein (46%); HP54—hemp press cake flour protein (54%); WFM—half-and-half mixture of wheat and whole wheat flour. Raw materials were measured in triplicate, and doughs and breads were measured nine times.

Sample	L*	a*	b*	ΔE*
HSG	55.6 ± 0.14	3.1 ± 0.02	12.7 ± 0.06	33.7 ± 0.12
HPF	60.0 ± 0.03	5.2 ± 0.01	17.9 ± 0.02	30.3 ± 0.01
HP46	55.2 + ±0.01	5.4 ± 0.03	18.7 ± 0.04	35.2 ± 0.03
HP54	73.0 ± 0.01	9.4 ± 0.01	25.2 ± 0.03	23.0 ± 0.02
WFM	89.2 ± 0.02	3.4 ± 0.01	10.0 ± 0.01	
HSG-1-D	71.5 ± 0.22	8.0 ± 0.08	18.3 ± 0.09	1.0 ± 0.43
HPF-1-D	68.8 ± 0.24	7.9 ± 0.09	19.3 ± 0.19	2.4 ± 0.40
HP46-1-D	69.2 ± 0.39	7.9 ± 0.16	19.3 ± 0.23	2.1 ± 0.27
HP54-1-D	70.9 ± 0.46	8.6 ± 0.06	19.5 ± 0.20	0.5 ± 0.20
HPF-18-D	51.4 ± 0.18	5.4 ± 0.12	17.0 ± 0.16	20.1 ± 0.32
HP46-14-D	54.2 ± 0.68	5.6 ± 0.17	19.1 ± 0.48	17.2 ± 0.35
HP54-10-D	67.2 ± 0.33	9.3 ± 0.04	23.8 ± 0.15	6.2 ± 0.46
WFM-D	71.1 ± 0.64	8.6 ± 0.12	19.0 ± 0.14	
HSG-1-B	64.7 ± 0.94	8.0 ± 0.03	17.3 ± 0.07	1.3 ± 0.80
HPF-1-B	59.3 ± 0.38	8.3 ± 0.07	17.9 ± 0.26	4.7 ± 0.24
HP46-1-B	59.7 ± 0.44	7.7 ± 0.13	17.3 ± 0.25	4.5 ± 0.31
HP54-1-B	63.0 ± 0.33	9.1 ± 0.19	19.3 ± 0.36	1.5 ± 0.30
HPF-18-B	43.6 ± 0.48	4.2 ± 0.07	8.3 ± 0.08	23.1 ± 0.34
HP46-14-B	47.6 ± 0.65	5.2 ± 0.07	11.7 ± 0.47	18.0 ± 0.58
HP54-10-B	56.9 ± 0.24	9.5 ± 0.43	20.0 ± 1.00	7.4 ± 0.92
WFM-B	64.0 ± 0.15	8.6 ± 0.05	18.2 ± 0.17	

**Table 3 molecules-27-01840-t003:** Mean results of texture profile analysis with standard deviations of doughs (D) and breads (B). HSG—hemp seed grit; HPF—hemp press cake flour; HP46—hemp press cake flour protein (46%); HP54—hemp press cake flour protein (54%); WFM—half-and-half mixture of wheat and whole wheat flour. All samples were measured at least nine times.

Sample	Stickiness /N	Hardness/N	Cohesion	Elasticity	Gumminess /N	Chewiness /N
	D	B	B	B	B	B
HSG-1	−0.67 ± 0.09 *	7.60 ± 0.67 *	0.97 ± 0.02 *	1.06 ± 0.01 *	7.39 ± 0.60 *	7.81 ± 0.67 *
HPF-1	−0.37 ± 0.12 *	8.79 ± 1.17 *	0.92 ± 0.03 *	1.06 ± 0.01 *	8.10 ± 1.13 *	8.57 ± 1.22 *
HP46-1	−0.54 ± 0.09	6.24 ± 0.75	0.96 ± 0.03 *	1.05 ± 0.01 *	5.99 ± 0.60 *	6.28 ± 0.61
HP54-1	−0.26 ± 0.10 *	13.45 ± 1.39 *	0.89 ± 0.02 *	1.06 ± 0.01 *	11.94 ± 1.21 *	12.67 ± 1.33 *
HPF-18	−0.69 ± 0.03 *	14.77 ± 1.18 *	0.90 ± 0.02 *	1.09 ± 0.01	13.25 ± 0.87 *	14.45 ± 1.00 *
HP46-14	−0.70 ± 0.02 *	15.71 ± 1.14 *	0.92 ± 0.02 *	1.10 ± 0.02 *	14.44 ± 1.15 *	15.91 ± 1.45 *
HP54-10	−0.42 ± 0.04 *	7.64 ± 0.99 *	1.07 ± 0.12	1.13 ± 0.02 *	8.12 ± 1.21 *	9.19 ± 1.46 *
WFM	−0.56 ± 0.04	5.84 ± 0.47	1.05 ± 0.0.2	1.08 ± 0.01	6.13 ± 0.57	6.63 ± 0.65

* Samples differ significantly from the reference (*p* < 0.05).

**Table 4 molecules-27-01840-t004:** Nutritional values of all raw materials according to nutrition labels. HSG—hemp seed grit; HPF—hemp press cake flour; HP46—hemp press cake flour protein (46%); HP54—hemp press cake flour protein (54%); WF—wheat flour; WWF—whole wheat flour.

Raw Material	Energy/kJ/100 g	Protein/g/100 g	Fat/g/100 g	Carbohydrates/g/100 g	Dietary Fibre/g/100 g
HSG	2380	28	46	4.8	14
HPF	1200	33	6.8	3.3	42
HP46	1374	46	8.7	1.6	31
HP54	1341	54	4.5	8.2	16
WF	1453	12	1.7	68	3.7
WWF	1422	13	3	58	13

## Data Availability

The data presented in this study are available on request from the corresponding author. The data are not publicly available because consultation with the funding agency and consortium partners is required for raw data publication.

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
