# Peer review of "Technological Changes in Wheat-Based Breads Enriched with Hemp Seed Press Cakes and Hemp Seed Grit"

_molecules, 2022, doi:10.3390/molecules27061840_

Round 1

Reviewer 1 Report

This work presents a series of investigations regarding the characteristics of some varieties of bread with the addition of hemp seed press cakes and hemp seed grit. There are several points which needs to be seriously taken care off and then the manuscript could be considered for its publication. 

  1. ”Principle component analysis” - must be replaced with the principal component analysis entire the manuscript;
  2. Keywords need to be reviewed, for example ”ingredient substitution” cannot be a keyword;
  3. ”Consequently, while various hemp materials present with great nutritional value, a balance between nutritional enhancement and bread quality and consumer acceptance has to be met.”This statement is not supported by the results obtained;
  4. Regarding the acceptance of this type of bread by consumers, the authors did not make any study, so the statement cannot be taken into account;
  5. The introduction needs to be improved, the novelty of the paper needs to be clarified and the text needs to be supplemented with more bibliographical references;
  6. Figures 2 and 3 need to be redone for clarity and understanding;
  7. Table 4. - the abbreviations for the raw materials specified in the table do not correspond to the explanations above;
  8. How was the protein content determined in the obtained bread assortments ? 

Author Response

Dear madam/sir,

thank you very much for all your comments and recommendation to our manuscript. Regarding to our recommendation see above our answers and adations.

  • Principle component analysis” - must be replaced with the principal component analysis entire the manuscript;

A: “Principal component analysis” was corrected in every instance.

  • Keywords need to be reviewed, for example ”ingredient substitution” cannot be a keyword;

A: “Ingredient substitution” was removed. The keyword “hemp bread” was split into two keywords: “hemp” and “bread”. Additionally, the key word “nutritional food quality” was removed and “by-product up-cycling” was added as a keyword.

  • ”Consequently, while various hemp materials present with great nutritional value, a balance between nutritional enhancement and bread quality and consumer acceptance has to be met.” This statement is not supported by the results obtained;

A: The sentence was revised to “Consequently, while various hemp materials present with great nutritional value, a balance with sensory properties, e.g. textural and color, has to be met.

  • Regarding the acceptance of this type of bread by consumers, the authors did not make any study, so the statement cannot be taken into account;

A: The respective sentences and paragraphs were revised. This includes revisions in abstract, introduction, discussion and conclusion.

  • The introduction needs to be improved, the novelty of the paper needs to be clarified and the text needs to be supplemented with more bibliographical references;

A: The introduction was revised. It now includes a paragraph highlighting other relevant studies and the aim and novelty of the presented study is now presented more clearly. Eight bibliographical references were added: https://doi.org/10.3390/pr9101782, https://doi.org/10.3390/molecules26154641, https://doi.org/10.3390/foods9091192, https://doi.org/10.3390/foods10040819, https://doi.org/10.1016/j.lwt.2018.01.029, https://doi.org/10.1002/cche.10043, https://doi.org/10.1016/j.jfoodeng.2010.07.017, https://doi.org/10.3390/plants10081558.

  • Figures 2 and 3 need to be redone for clarity and understanding;

A: Figures 1, 2 and 3 were revised.

  • Table 4. - the abbreviations for the raw materials specified in the table do not correspond to the explanations above;

A: The table caption was revised.

  • How was the protein content determined in the obtained bread assortments?

A: As stated in the introduction, the EU health claim high protein is defined as 20 % of energy coming from protein. The necessary amounts to reach this claim were calculated according to the information provided by the supplier of the raw materials. This information is now presented more clearly in the introduction and materials & methods.

Warm regards,

Dr. Katrin Bach

Reviewer 2 Report

Recommendations: 1. It is advisable to describe abbreviations before using them (for example, DM is used immediately in the ''Results'' section without prior explanation, and it is explained later in the table description). 2. For a more accurate explanation of the results of the distribution of the raw material by size, it is recommended to specify the width of the gap or the level of blackout as well as the indication of the feed rate of the raw material. 3. There are no recommendations about the rational dosing of hemp raw materials based on research. 4. It is recommended to give indicators of quality of finished bakery products with added raw materials. 5. It is recommended to sign the abscissa in Figure 2b. Figure 3 is not divided into 3a and 3b.

Author Response

Dear madam/sir,

thank you very much for all your comments and recommendation to our manuscript. Regarding to our recommendation see above our answers and adations:

  1. It is advisable to describe abbreviations before using them (for example, DM is used immediately in the ''Results'' section without prior explanation, and it is explained later in the table description).

A: Abbreviations are now described before first use in every instance.

  1. For a more accurate explanation of the results of the distribution of the raw material by size, it is recommended to specify the width of the gap or the level of blackout as well as the indication of the feed rate of the raw material.

A: The feed rate and pressure as well as the instrument model are now not only listed in the methods & materials sections, but also in figure caption 1 and text in section 2.1.

  1. There are no recommendations about the rational dosing of hemp raw materials based on research.

A: An explanation why 1 % was chosen was added in the introduction. The higher dosages were chosen to reach the EU health claim high protein. Therefore, the necessary amounts of each hemp raw material for reaching this claim were calculated according to the nutritional information provided by the supplier of raw material. This is now explained in more detail in the introduction and materials & methods.

  1. It is recommended to give indicators of quality of finished bakery products with added raw materials.

A: Relevant quality parameters are summarized in table 3. The focus of the presented study was on industrial bread quality and the feasibility of high protein breads in an industrial context. Consequently, in this first step the main focus was on the rheology as an indicator for a technological application of the products. This is now also stated in the introduction.

  1. It is recommended to sign the abscissa in Figure 2b. Figure 3 is not divided into 3a and 3b.

A: The abscissa now has an axis title. Figure 3 was divided into a & b.

Warm regards,

Dr. Katrin Bach

Reviewer 3 Report

line 18 – Mastersizer is a device for measuring particle sizes, produced by certain company. So I recommend to give model number and company name or delete the end part of the sentence “…using a mastersizer”.

The authors should add a paragraph describing recent developments in supplementing wheat breads with other non-gluten additives in order to improve texture and nutritional quality.  Here I recommend the literature in this topic:

https://doi.org/10.3390/pr9101782

https://doi.org/10.3390/molecules26154641

https://doi.org/10.3390/foods9091192

https://doi.org/10.3390/foods10040819

Figure 1 - figure has no description divided into a and b parts.

Figure 2 is completely unreadable. Parameter abbreviations and graphics representing individual variants and parameters are so small that the picture does not convey any information. Please enlarge all letters and characters on figures as well as descriptions of both axes.

Table2 and 3 – authors in the text mention very often the significant or not significant differences between samples but there is no statistics provided in table 2 and 3. Statistics must appear in these tables!

Figure 3- same comment as for figure 2

line 260 – correct citation

line 322-323 and Table 4 – due to my and my colleagues experience nutritional values given on raw materials labels differs from real composition so in scientific paper these labeled values are not reliable and cannot be used. Authors rely on nutritional composition of raw materials (figure 2) in their PCA analysis so if they want to keep it and use it than authors should perform analyses to obtain real composition of raw hemp materials.

lines 334-346 – the ingredient composition of each variant doughs is better to present using a table. Visualization of percentage changes of certain ingredients like flour, hemp materials, water will allow a clearer presentation of the changes in each doughs variant to the reader. I highly recommend implementation of this change.

texture analysis – please describe the kind of spindle or probe used for analyses

line 410-412 – please describe better with more details the principal component analysis (PCA). Are all analyzed parameters qualified for PCA analysis? What was the lowest correlation value of parameters with the generated first and second principal components?

Author Response

Dear madam/sir,

thank you very much for all your comments and recommendation to our manuscript. Regarding to our recommendation see above our answers and adations.

  1. line 18 – Mastersizer is a device for measuring particle sizes, produced by certain company. So I recommend to give model number and company name or delete the end part of the sentence “…using a mastersizer”.

A: The sentence ending was deleted.

  1. The authors should add a paragraph describing recent developments in supplementing wheat breads with other non-gluten additives in order to improve texture and nutritional quality.  Here I recommend the literature in this topic: https://doi.org/10.3390/pr9101782, https://doi.org/10.3390/molecules26154641, https://doi.org/10.3390/foods9091192, https://doi.org/10.3390/foods10040819

A: The introduction was revised and a paragraph discussing literature in more detail was added. In addition to the recommended sources the following sources were added as well: https://doi.org/10.1016/j.lwt.2018.01.029, https://doi.org/10.1002/cche.10043, https://doi.org/10.1016/j.jfoodeng.2010.07.017, https://doi.org/10.3390/plants10081558.

  1. Figure 1 - figure has no description divided into a and b parts.

A: The figure caption was revised and now includes parts a & b.

  1. Figure 2 is completely unreadable. Parameter abbreviations and graphics representing individual variants and parameters are so small that the picture does not convey any information. Please enlarge all letters and characters on figures as well as descriptions of both axes.

A: Figure 2 was revised.

  1. Table2 and 3 – authors in the text mention very often the significant or not significant differences between samples but there is no statistics provided in table 2 and 3. Statistics must appear in these tables!

A: Thank you for your comment. Table 2 already provided the standard deviation for all L*, a* and b*. The standard deviation for ΔE* was added. Color differences are commonly only discussed regarding to their ΔE* value, especially when dealing with foods, consequently no additional statistical evaluation was added in the table. For table 3, the statistical results are now shown in the table. Statistics is now also presented in table 1.

  1. Figure 3- same comment as for figure 2

A: The figure was revised.

  1. line 260 – correct citation

A: The citation was corrected.

  1. line 322-323 and Table 4 – due to my and my colleagues experience nutritional values given on raw materials labels differs from real composition so in scientific paper these labeled values are not reliable and cannot be used. Authors rely on nutritional composition of raw materials (figure 2) in their PCA analysis so if they want to keep it and use it than authors should perform analyses to obtain real composition of raw hemp materials.

A: Thank you for your comment. The true nutritional values truly often differ from the values given on the label due to biological and technical variations (e.g. charge depended variations). However, these discrepancies are usually in a narrow range and considering the great differences between raw materials (see table 4), it is highly unlikely that a more accurate assessment of the nutritional value would change the results. Furthermore, as PCA is a qualitative assessment method to get a relative understanding about similarities between samples we believe that the great laboratory effort required to generate the data would be disproportionate to the likely small gain. Lastly, due to time passing since the study, the exact dough and bread samples are no longer available for analysis and preparing new doughs and breads and analyzing them is error prone as well.

  1. lines 334-346 – the ingredient composition of each variant doughs is better to present using a table. Visualization of percentage changes of certain ingredients like flour, hemp materials, water will allow a clearer presentation of the changes in each doughs variant to the reader. I highly recommend implementation of this change.

A: The ingredient composition is now displayed as a table.

  1. texture analysis – please describe the kind of spindle or probe used for analyses

A: The type of probe used is now listed in section 4.4.

  1. line 410-412 – please describe better with more details the principal component analysis (PCA). Are all analyzed parameters qualified for PCA analysis? What was the lowest correlation value of parameters with the generated first and second principal components?

A: All parameters were scaled to unit vectors, in order ensure that the data set is suitable for PCA. This information was added in the results section as well as in figure captions 2 and 3. Furthermore, screeplots helped to gain an understanding about the data reduction potential and since at least two components yielded an Eigenvalue below 1, PCA was shown to be suitable data evaluation tool. The two screeplots for figure 3 were added as a supplementary figure. The loadings are now discussed in the results section are provided as a supplementary table 4.

Warm regards,

Katrin Bach

Round 2

Reviewer 1 Report

The authors reviewed the manuscript, some of the observations were resolved. But the following issues remain:

- Principle component analysis” - must be replaced with the principal component analysis entire the manuscript-the authors need to re-check and modify the text in the figures as well;

-Figures 2 and 3 need to be redone for clarity and understanding; here the main problems are the size of the writing and the clarity of the images

-regarding the protein content, it is necessary to explain in detail how the protein content was determined (calculated) in the obtained dough and bread assortments.

Author Response

Dear Sir/Madam,

thank you very much for your comments. We answered your question down below.

- Principle component analysis” - must be replaced with the principal component analysis entire the manuscript-the authors need to re-check and modify the text in the figures as well;

A: “Principle component analysis” was replaced with “principal component analysis” in every instance (text & figures)

-Figures 2 and 3 need to be redone for clarity and understanding; here the main problems are the size of the writing and the clarity of the images.

A: Figures 2 & 3 were revised again. The text was enlarged. Regarding clarity: It seems that the mdpi server greatly reduces figure quality in order to produce a small file to send to reviewers. The original figures are at 300 dpi (and have always been) and thus blurring is not an issue. Since mdpi will use the original figures for publication, there should be no problem with the clarity of the images in the finished manuscript. We can provide the reviewer with our original jpg-files to prove the sufficient image quality.

-regarding the protein content, it is necessary to explain in detail how the protein content was determined (calculated) in the obtained dough and bread assortments.

A: The equation how protein content was calculated was added in the method and material section.

Reviewer 3 Report

The authors have made all recommended corrections or addressed the comments. The article has been greatly improved and now is more readable. I have no further comments on the manuscript.

Author Response

Dear Sir/Madam,

Thank you very much for your comments. An additional spelling and grammar correction were done.

Kind regards,

K. Bach